# FLOW-BASED ALIGNMENT OF UNI-MODAL VISION AND TEXT ENCODERS FOR FEW-SHOT IMAGE CLASSIFICATION

## ABSTRACT

Few-shot classification with vision–language models remains challenging, particularly when relying on multi-modal encoders such as CLIP that are restricted to paired image–text data. We introduce FSF, a framework that leverages arbitrary uni-modal encoders—including vision or text models that were pretrained on broad or domain-specific corpora—and aligns them for cross-modal classification. FSF first applies a closed-form orthogonal Procrustes map to align image and text embeddings while preserving their geometry, and then trains a lightweight flow-matching prior that regularizes adaptation in the few-shot regime. At inference, images are classified by cosine similarity in the aligned feature space between query embeddings and mapped class prototypes. Experiments on standard benchmarks, ImageNet variants, and VinDr-CXR, a large-scale chest X-ray benchmark, show that FSF is able to leverage stronger or specialized encoders, achieving competitive or superior accuracy compared to recent adaptation methods.

## 1 INTRODUCTION

Few-shot image classification (FSC) has become an important benchmark for evaluating how well models can adapt to new tasks with minimal supervision. Unlike standard classification, FSC stresses the ability to generalize from only one or a handful of labeled examples per class, often in settings that are out-of-distribution with respect to the pretraining data.

The availability of large *multi-modal* (jointly trained) vision–language encoders, most prominently CLIP (Radford et al., 2021), has reshaped this landscape: images and class names are embedded into a shared representation space, so classification reduces to comparing image features against text-prompt prototypes. This simple mechanism has proven so effective that CLIP-based methods have largely displaced traditional vision encoders as the backbone for FSC.

On top of multi-modal embedding spaces such as CLIP, a wide range of *adapters* has been proposed to push FSC further. Prompt-tuning methods (Zhou et al., 2022) optimize class descriptions, cache-based calibration (Zhang et al., 2022) reuses support examples, and feature- or logit-level adapters (Gao et al., 2021; Zhang et al., 2023) introduce small modules to adjust the embedding space. These methods achieve strong results because they exploit the geometry already embedded by joint pretraining. Yet this reliance is limiting: performance depends on the quality of the original multi-modal training and the use of uni-modal encoders that may provide stronger or more domain-specialized representations is prevented.

A different line of work instead focuses on leveraging *uni-modal* (separately trained) encoders, connecting independent vision and text models by learning projectors between their latent spaces. This demonstrates that frozen uni-modal encoders can be bridged in a post-training manner, but existing methods (Zhai et al., 2022; Li et al., 2023a; Alayrac et al., 2022) are aimed at general multi-modal tasks—zero-shot recognition, VQA, or cross-modal retrieval and generation—rather than FSC. They typically require large paired datasets and heavy adapter training, making them poorly suited to the few-shot setting.

Another approach is to align vision and language embeddings with *linear maps*. Some methods learn these maps from data (Mikolov et al., 2013; Frome et al., 2013), while others fit them di-

rectly (Merullo et al., 2023). This line of work shows that direct and meaningful cross-modal correspondence can emerge from such mappings. This simplicity makes linear alignment appealing, particularly since in some cases it can be fitted without training. However, purely-linear mappings cannot capture the non-linear dependencies between modalities that become important for robust adaptation (Sung et al., 2022; Chen et al., 2023; Li et al., 2023c).

More recently, continuous-time generative models such as *diffusion* (Ho et al., 2020) and *rectified flow* (Liu et al., 2023; Lipman et al., 2023) have shown how non-linear dependencies can be modeled through smooth transformations. While much of this work has focused on pixel-level generation, subsequent advances (Wu et al., 2024; Zhang et al., 2025b;c) demonstrated that such flows can also operate in vision–language latent spaces. This suggests a way to capture richer cross-modal structure beyond what linear mappings allow, motivating their adaptation to FSC where supervision is scarce.

Our approach builds on two complementary components. The first is an *Orthogonal Procrustes* linear alignment (Schönemann, 1966), which serves as a preprocessing step that harmonizes independently trained embeddings, potentially of different feature dimensionalities, into a shared coordinate system. The second is a lightweight *flow-matching prior* (Lipman et al., 2023; Liu et al., 2023) that we apply directly in this aligned latent space. By modeling cross-modal transport as a time-continuous process, the flow introduces the expressive capacity needed for moving beyond linear mappings while being efficient enough for the few-shot regime.

The resulting framework, **FSF** (Few-Shot-Flow), combines these two components, as can be seen in the workflow presented in Fig. 1. Orthogonal Procrustes provides a stable initialization, over which the flow operates to learn velocity fields along simple paths—linear or geodesic on the unit sphere—between text prototypes and image features. At inference time, bidirectional flows are used to integrate text and image latents toward intermediate representations, enabling similarity-based matching between images and classes. This combination maintains the efficiency of closed-form linear alignment while adding the flexibility of latent flows.

Extensive experiments demonstrate the value of FSF for flexible alignment of uni-modal encoders. First, across independently trained image–text combinations, FSF consistently outperforms equivalent baselines, showing clear benefits from aligning different pairs of uni-modal encoders. On ImageNet distribution-shift benchmarks, FSF is competitive in the multi-modal setting, and with CLIP text + DINO vision it achieves excellent results—surpassing comparable multi-modal variants and even a DINO linear probe on the target sets. In the medical domain, on the VinDr-CXR chest X-ray benchmark (Nguyen et al., 2020), generic CLIP and DINO models perform poorly, whereas FSF leverages the RAD-DINO vision encoder (Pérez-García et al., 2024) with either large or domain-specific text models to achieve strong results, surpassing RAD-DINO linear probing with only 128 shots per class. Finally, across 11 few-shot benchmarks, FSF is competitive with the standard CLIP-RN50 backbone and delivers consistent gains when substituting the CLIP vision encoder with a DinoV2-S encoder.

## 2 RELATED WORK

### 2.1 FEW-SHOT ADAPTERS FOR JOINT (MULTI-MODAL) VISION–LANGUAGE MODELS

A major line of work adapts *joint* vision–language models, most prominently CLIP Radford et al. (2021), to downstream tasks using few-shot supervision. These approaches assume a co-trained image–text embedding space and introduce parameter-efficient modules while keeping the backbones frozen. Training-free and cache-based methods such as TIP-Adapter Zhang et al. (2022) and SuS-X Udandarao et al. (2023) avoid gradient updates by constructing a support-set cache and combining retrieval-based similarity with the original CLIP logits. Prompt-learning approaches, exemplified by CoOp Zhou et al. (2022), optimize learnable text prompts or biases, thereby reshaping the textual prototypes used for classification.

Another prominent direction adapts models by modifying either their intermediate representations or their decision scores. In the feature space, methods such as CLIP-Adapter Gao et al. (2021), CaFo Zhang et al. (2023), APE Zhu et al. (2023), DMN Zhang et al. (2024), and CLAP Silva-Rodríguez et al. (2024) attach lightweight, parameter-efficient modules that adjust embeddings under few-shot supervision. In the logit space, approaches such as AWT Zhu et al. (2024) refine the

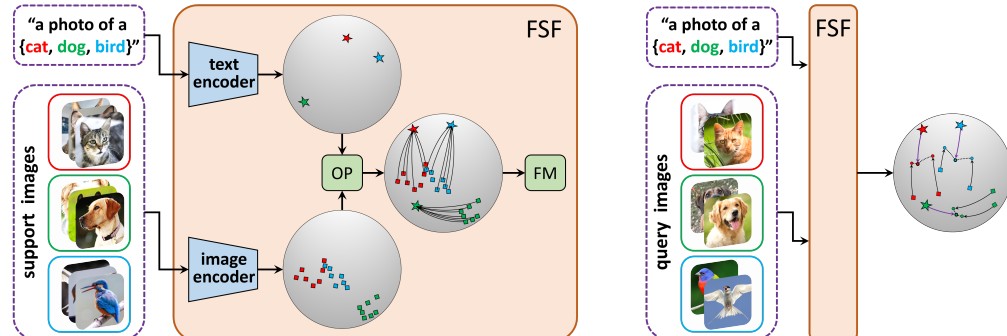

Figure 1: **FSF workflow**. *Training* (**left**): Support images and class labels are passed through frozen image and text encoders, producing unit-normalized features that are not well aligned. Orthogonal Procrustes (OP) alignment (Sec. 3.2) applies a closed-form linear map to the text features, preserving their relative geometry and providing initialization for flow-based alignment. A lightweight MLP flow-matching (FM) module (Sec. 3.3) is then trained to map text latents to image latents along rectified linear or geodesic paths on the unit sphere. *Inference* (**right**): Query images are processed by the same FSF modules. Image and text latents are integrated along forward and reverse flows to intermediate points, shown here as circles. Distances (dashed lines) between these points serve as similarities for argmax-style class prediction. See Fig. 2 for further details.

class scores through calibration or bias. Structure-aware extensions such as GraphAdapter Li et al. (2023d) further enrich this family with relational perspectives.

Together, these methods illustrate the spectrum of parameter-efficient fine-tuning strategies for co-trained VLMs. In contrast, FSF does not rely on a jointly pretrained multi-modal space: it accepts independently trained encoders and aligns them through a closed-form orthogonal Procrustes map, before stabilizing few-shot adaptation with a lightweight, parameter-efficient flow-matching prior.

## 2.2 ALIGNMENT OF INDEPENDENT (UNI-MODAL) ENCODERS

Another line of work explores bridging independently pretrained image and text encoders. LiT Zhai et al. (2022), for instance, learns a text projection on top of a frozen vision model using paired image–text data, while BLIP-2 Li et al. (2023a) introduces a Q-Former to connect vision features with a large language model. Freeze-Align Zhang et al. (2025a) follows a similar philosophy, training a projector on external paired corpora to flexibly align frozen uni-modal encoders, but requires medium-scale training resources. Although effective, these approaches depend on large paired corpora and substantial optimization.

Closer to our setting are methods that pursue linear alignment. LiMBeR Merullo et al. (2023) learns a linear projection from image embeddings into a language model's space, trained on external data and evaluated in zero-shot generation tasks such as captioning or VQA. LFA Wu et al. (2023) is parameter-free, applying an orthogonal Procrustes solution to align embeddings across modalities, and thus demonstrates that simple closed-form mappings can already be competitive.

FSF adopts a similar orthogonal Procrustes alignment as a baseline step, but its main contribution lies in introducing a lightweight, parameter-efficient flow-matching prior that regularizes the aligned space. This additional component yields consistent improvements in few-shot adaptation, enabling FSF to leverage arbitrary uni-modal encoders without requiring external paired data.

## 2.3 LATENT CONTINUOUS-TIME MODELS FOR CROSS-MODAL GENERATION

Modern cross-modal generation systems encode the source input (e.g., text) into a latent space, then transform it into the latent space of the target domain, and finally decode the result back into the output modality (e.g., image).

Most approaches implement this transformation using continuous-time generative models, either diffusion or rectified flow, formulated as conditional processes from a noise prior to target latents guided by the source representation. DALL·E v2 Ramesh et al. (2022) encodes text with CLIP

and conditions a denoising diffusion process that maps noise to image latents, which are decoded with an autoencoder. Stable Diffusion v2 Rombach et al. (2022) follows the same paradigm in the latent space of a pretrained VAE, where diffusion proceeds from noise to image latents under text conditioning. Wu et al. (2024) replaces diffusion with rectified flow but retains the conditional formulation, training a transformer denoiser on Gaussian-to-image-latent flows with text guidance. Cross-Flow Zhang et al. (2025b) similarly flows from a Normal prior to image latents, while jointly optimizing a text encoder with a contrastive loss. In contrast, OmniFlow Zhang et al. (2025c) departs from this conditional pattern by directly learning flows between modality-specific latent spaces, supporting any-to-any (e.g. audio↔video) transfers.

While FSF is inspired by the way these methods perform continuous-time matching between multi-modal latent spaces, it addresses a different setting: non-generative few-shot adaptation. In this case, the transformation is a lightweight prior applied directly in feature space, without reliance on paired encoders–decoders or large-scale generative training.

## 3 METHOD

The proposed FSF is a modular adaptation framework for few-shot image classification, whose workflow is presented in Fig. 1. FSF (i) decouples the choice of image and text encoders (multi- or uni-modal); (ii) aligns them via a *training-free* closed-form Orthogonal Procrustes (OP) mapping; (iii) learns a lightweight yet expressive *flow-matching* prior in embedding space; and (iv) enables efficient inference. The OP map preserves within-modality structure while enabling cross-modal comparison, and the flow objective learns an ODE velocity field between image and text embeddings, yielding a regularized adaptation from very few shots. Training and inference procedures are summarized in Algorithms 1 and 2 (Appendix A.1).

### 3.1 PROBLEM SETUP AND NOTATIONS

In the few-shot classification setting, we are given a *support* set $\Omega = (\mathcal{S}, \mathcal{L})$ that consists of $N$ images $\mathcal{S} = \{s_i\}_{i=1}^N$, with corresponding labels $\mathcal{L} = \{l_i\}_{i=1}^N$ that belong to one of $C$ classes, with $K$ examples (shots) per class. A model can be trained on this supervised data and is tested at inference on a set test set of *query* images that belong to the same set of classes.

Let $f_{\text{img}}$ and $f_{\text{txt}}$ denote a pair of pretrained image and text encoders, with frozen parameters, that produce outputs of dimension $D_{\text{img}}$ and $D_{\text{txt}}$, respectively. In this work, we consider both the case of *multi-modal* encoders, which are jointly trained (with CLIP Radford et al. (2021) being the most common example), and *uni-modal* encoders, which are pretrained independently on domain-specific data (e.g., DINO Oquab et al. (2024) for vision and BERT Devlin et al. (2019) for language).

Each labeled sample $(s, l)$ is mapped into both image and text feature spaces. The resulting embeddings are normalized onto their respective unit spheres by

$$x = \frac{f_{\text{img}}(s)}{\|f_{\text{img}}(s)\|} \quad \text{and} \quad y = \frac{f_{\text{txt}}(\hat{l})}{\|f_{\text{txt}}(\hat{l})\|} \tag{1}$$

where $\hat{l}$ denotes either the raw class label $l$ (used as a prototype) or a fixed template sentence containing it.

### 3.2 TRAINING-FREE ALIGNMENT VIA ORTHOGONAL PROCRUSTES

We align the two embedding spaces with a *semi-orthogonal* linear map $W$, which is estimated from paired image–text data (which may be the support set or any other external source of paired data). The linear map is obtained with a closed-form solution of an Orthogonal Procrustes (OP) problem, at the cost of computing an SVD of a $D_{\text{txt}} \times D_{\text{img}}$ matrix. Being (semi-)orthogonal, the transformation preserves inner products in the source subspace and thus conserves within-modality relations of the source domain, which is desirable for achieving good alignment.

Given $n$ paired embeddings $\{(x_i, y_i)\}_{i=1}^n$ with $x_i \in \mathbb{R}^{D_{\text{img}}}$ and $y_i \in \mathbb{R}^{D_{\text{txt}}}$, we first stack them into matrices

$$\mathbf{X} = [x_1, \ldots, x_n]^\top \in \mathbb{R}^{n \times D_{\text{img}}}, \qquad \mathbf{T} = [y_1, \ldots, y_n]^\top \in \mathbb{R}^{n \times D_{\text{txt}}}.$$

To align the text and image feature spaces, we then solve an Orthogonal Procrustes (OP) problem

$$\mathbf{W}^\star = \underset{\mathbf{W} \in \mathbb{R}^{D_{\text{img}} \times D_{\text{txt}}}}{\arg \min} \left\| \mathbf{T} \mathbf{W}^\top - \mathbf{X} \right\|_F^2 \quad \text{s.t.} \quad \mathbf{W} \mathbf{W}^\top = \mathbf{I}_{D_{\text{img}}}. \tag{2}$$

Let $\mathbf{M} = \mathbf{X}^\top \mathbf{T}$ denote the cross-covariance matrix, and take its truncated SVD given by $\mathbf{M} = \mathbf{V} \mathbf{\Sigma} \mathbf{U}^\top$ with $\mathbf{U} \in \mathbb{R}^{D_{\text{txt}} \times r}$, $\mathbf{V} \in \mathbb{R}^{D_{\text{img}} \times r}$, where $r = \text{rank}(\mathbf{M})$.

The closed-form OP solution yields a $D_{\text{img}} \times D_{\text{txt}}$ mapping that is orthogonal when $D_{\text{img}} = D_{\text{txt}}$ (and otherwise row-orthonormal via the thin SVD), given by

$$\mathbf{W}^\star = \mathbf{V} \mathbf{U}^\top. \tag{3}$$

Finally, text embeddings are mapped into image feature space while image embeddings stay fixed:

$$\boldsymbol{x}_i \leftarrow \boldsymbol{x}_i \in \mathbb{R}^{D_{\text{img}}}, \qquad \boldsymbol{y}_i \leftarrow \mathbf{W}^\star \boldsymbol{y}_i \in \mathbb{R}^{D_{\text{img}}}. \tag{4}$$

Regarding the paired data that the mapping is fitted to, we consider two different options, which we denote by 'local OP' and 'global OP'. In the local setting, following LFA Wu et al. (2023), we use the entire training set of the few-shot task (e.g. 808 pairs for 8-shot classification on Caltech101), while in the global setting, we use 1M randomly sampled text-image pairs from the CC3M Sharma et al. (2018) dataset. There is a clear tradeoff between the two. The 'local' approach is better tailored to the particular domain, but may be inferior, in the low-shot regime. We compare these variants in the experiments.

### 3.3 FLOW-MATCHING PRIOR IN THE ALIGNED SPACE

We next learn a continuous-time velocity field that transports image embeddings toward their corresponding OP-aligned class text prototypes. For a sample pair $(\boldsymbol{s}, \boldsymbol{l})$, let $(\boldsymbol{x}, \boldsymbol{y})$ be the feature embeddings (Eq. 1) that have gone through OP-alignment (Eq. 4) - both residing in the common image-space of dimension $\mathbb{R}^{D_{\text{img}}}$. For simplifying the presentation, we abuse notation and write $(\boldsymbol{x}, \boldsymbol{y}) \sim \Omega$, to denote a pair of embedded features taken from the Support set.

We now consider a continuous path $\gamma(t) = \gamma(t; \boldsymbol{x}, \boldsymbol{y})$, with time parameter $t \in [0, 1]$, that provides a direct interpolation between $\boldsymbol{x}$ and $\boldsymbol{y}$ at a time varying velocity $u(t) = u(t; \boldsymbol{x}, \boldsymbol{y})$. Two useful instantiations of such a path are:

- **linear (Euclidean):** $\gamma(t) = (1 - t)x + ty$, with ground-truth velocity $u(t) = \dot{\gamma}(t) = y - x$.

- **geodesic (on sphere):** $\gamma(t) = \text{slerp}(x, y; t)$ with analytic tangent velocity $u(t) = \dot{\gamma}(t)$. [1]

The geodesic choice (Chen & Lipman, 2023) respects spherical geometry and we found that it performs consistently better, as we show in Sec. of the appendix.

Following common practice in flow matching Lipman et al. (2023); Liu et al. (2023), we parameterize a time-conditioned velocity field $v_\theta : [0, 1] \times \mathbb{R}^{D_{\text{img}}} \to \mathbb{R}^{D_{\text{img}}}$ and train it to match the target velocity along the path. Sampling $t \sim \mathcal{U}(0, 1)$ and setting $\boldsymbol{x}_t = \gamma(t; \boldsymbol{x}, \boldsymbol{y})$, the *flow-matching* loss (in the image→text direction) is

$$\mathcal{L}_\rightarrow(\theta) = \mathbb{E}_{(x,y) \sim \Omega, \, t \sim \mathcal{U}[0,1]} \left[ \| v_\theta(t, \boldsymbol{x}_t) - \boldsymbol{u}(t; \boldsymbol{x}, \boldsymbol{y}) \|_2^2 \right]. \tag{5}$$

Our approach uses bi-directional flow in order to obtain better classification performance. Since we are using rectified flows (or ones that follow geodesic lines), one possibility is to model a single velocity field and integrate along it in the reverse direction for text to image mapping. However, we tested with training an identical additional model $v_{\theta'}$ (with an independent parameter set), using the equivalent loss $\mathcal{L}_\leftarrow$ that switches the roles of $\boldsymbol{x}$ and $\boldsymbol{y}$:

$$\mathcal{L}_\leftarrow(\theta') = \mathbb{E}_{(x,y) \sim \Omega, \, t \sim \mathcal{U}[0,1]} \left[ \| v_{\theta'}(t, \boldsymbol{y}_t) - \boldsymbol{u}(t; \boldsymbol{y}, \boldsymbol{x}) \|_2^2 \right]. \tag{6}$$

In practice, we found the option of using two independent flow fields to work slightly (but consistently) better than using a single model and its reverse velocity. We attribute this to the regularization effect of model ensembeling, given by the independent initialization and time sampling of the flow netowrks.

---

[1] $\text{slerp}(\boldsymbol{x}, \boldsymbol{y}; t) = \frac{\sin((1-t)\theta)}{\sin \theta} \boldsymbol{x} + \frac{\sin(t\theta)}{\sin \theta} \boldsymbol{y}$, $\dot{\gamma}(t) = \frac{-\cos((1-t)\theta)\,\theta}{\sin \theta} \boldsymbol{x} + \frac{\cos(t\theta)\,\theta}{\sin \theta} \boldsymbol{y}$, for $\theta = \arccos(\boldsymbol{x}^\top \boldsymbol{y})$.

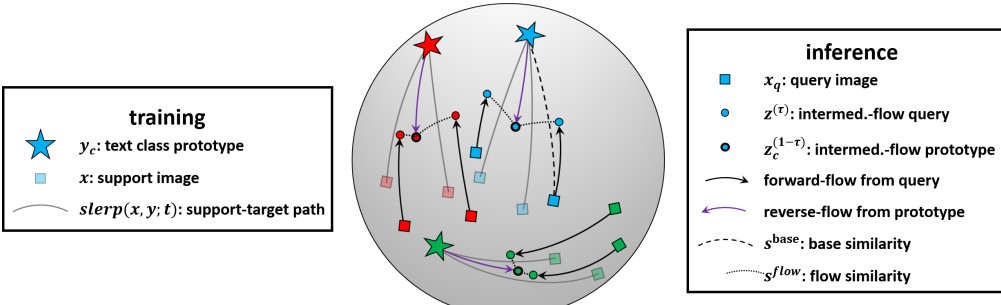

Figure 2: **Detail on flow-based inference**. At training, geodesic paths (grey solid lines) between class prototypes (stars) and corresponding support features (semi-transparent squares) are used to train forward and reverse flow fields in the joint latent space. At inference, unlabeled query images and class prototypes are integrated up to an intermediate timestep $\tau \in [0, 1]$ (here $\tau \approx 0.5$). The resulting intermediate points are shown as circles, with bold circles marking prototypes' locations. Classification uses a weighted combination of flow similarity (fine dashed curves) and base similarity (coarse dashed curves), with the predicted class given by the argmax of the resulting scores.

### 3.4 INFERENCE (CLASSIFICATION)

At test time, for any given unlabeled sample $s$, we compute its unit-normalized image feature $x$ and compute the OP-projected class prototypes $y_c \leftarrow W^\star y_c$ for each class $c \in C$.

Inference is done in the joint latent space, on the unit-sphere of the image space $\mathbb{R}^{D_{\text{img}}}$. Please follow the illustration in Fig. 2 for a detailed visualization of the inference process on the sphere.

We proceed by using the ODE defined by the learned flow field $v_\theta$

$$\frac{d\boldsymbol{z}}{dt} = v_\theta(t, \boldsymbol{z}), \qquad \boldsymbol{z}(0) = \boldsymbol{x}, \tag{7}$$

to propagate the image feature $x$ to a predetermined intermediate time $\tau \in [0, 1]$ (which we treat as a hyper-parameter). This is done by integration, using an adaptive solver (e.g., `dopri5`), and we denote the resulting feature point, which we unit-normalize, by $\boldsymbol{z}^{(\tau)}$.

Symmetrically, we transport each of text prototypes in the opposite direction, from the initial position of $\boldsymbol{z}_c(0) = \boldsymbol{y}_c$ to a resulting point at the matching intermediate time $1 - \tau$, which we unit normalize and denote by $\boldsymbol{z}_c^{(1-\tau)}$.

Class association, per query image, is taken to be the one which minimizes the cosine similarity between the flow-transported latents over all possible classes $c \in C$

$$s_c^{\text{flow}} = \left\langle \boldsymbol{z}^{(\tau)}, \boldsymbol{z}_c^{(1-\tau)} \right\rangle. \tag{8}$$

Following prior works (Gao et al., 2021; Zhang et al., 2023) we mix between the flow-transported scores $s_c^{\text{flow}}$ and the respective base (pre-flow) similarly scores $s_c^{\text{base}} = \left\langle \boldsymbol{x}, \boldsymbol{z}_c \right\rangle$, using a single scalar hyper-parameter $\alpha \in [0, 1]$:

$$s_c = (1 - \alpha)\, s^{\text{flow}} + \alpha\, s^{\text{base}}. \tag{9}$$

In practice, as we demonstrate in an ablation study (Sec. A.3.2) we pick the intermediate time parameter $\tau$ and mixing parameter $\alpha$ using the task-respective metric (accuracy for few-shot classification or macro-AUPRC in the multi-label setting) on a small subset of the task validation set.

**Discussion.** OP is used *once* to harmonize spaces while preserving modality-internal structure. It does not by itself "boost" accuracy but provides a stable geometric substrate for the learned flow. The flow objective then learns smooth, deterministic transport, consistent with OP geometry, that exploits non-linear relationships between strong, pre-trained encoders. Furthermore, we observed that uniform sampling of $t$ regularizes training by exposing the model to the entire interpolation path, reducing overfitting even over long schedules.

Table 1: **Encoder usage flexibility on 16-shot ImageNet Classification.** The **left** panel presents performance of different methods, using multi-modal CLIP encoders (either RN50 or the larger ViT-B/16). These methods, except for LFA and FSF, are constrained to working with jointly trained models, like CLIP. In contrast, shown in the **right** panel, LFA and FSF have the flexibly to align a variety of independently pretrained encoders. FSF is shown to be consistently superior to LFA, in performing these adaptations. The bottom examples (below the dashed line) show how the use of strong/general uni-modal text (Qwen/GTE/ARL) and vision (DINOv2-B) encoders, can be extremely beneficial over using multi-modal CLIP at matched ViT-B scale (right column of left pane).

| method | CLIP-RN50 | CLIP-ViT-B/16 | | text encoder | vision encoder | LFA | **FSF**-LOP |
|---|---|---|---|---|---|---|---|
| ZS-CLIP | 60.33 | 68.73 | | CLIP-RN50 | MAE-B | 27.76 | 39.75 |
| Tip-Adapter | 61.43 | 70.25 | | CLIP-RN50 | BAM-B | 66.43 | 68.97 |
| CoOp | 62.95 | 71.92 | | CLIP-RN50 | DINOv2-B | 75.03 | 76.80 |
| CLIP-Adapter | 63.59 | 71.13 | | CLIP-ViT-B | DINOv2-B | 74.07 | 76.70 |
| ProKeR | *64.45* | *73.25* | | GTE-B | DINOv2-B | 73.57 | 75.77 |
| Tip-Adapter-F | **65.51** | **73.69** | | GTE-L | DINOv2-B | 73.27 | 75.90 |
| LFA | 63.65 | 72.61 | | ARL | DINOv2-B | 73.27 | 75.93 |
| **FSF**-noOP | 64.07 | *73.27* | | Qwen-8B | DINOv2-B | 74.84 | 76.37 |

## 4 EXPERIMENTS

### IMPLEMENTATION DETAILS

We evaluate three variants: **FSF**-LOP (local OP) and **FSF**-GOP (global OP) for uni-modal encoders, and **FSF (no-OP)** for CLIP-based multi-modal settings. In CLIP setups we omit OP because joint pretraining sufficiently aligns image–text latents. The velocity field $v_\theta(t, z)$ follows Li et al. (2023b): a residual MLP with SiLU activations and per-layer time conditioning. We use 4 layers with hidden width 1536 (input/output match encoder dimensions) and ablate other options in Sec. A.3.3. Training uses AdamW (lr $= 1e-4$, weight decay $= 1e-3$) with a cosine schedule. Encoders and the OP/GOP map are frozen. For inference, we transport query embeddings and class prototypes with an adaptive ODE solver (dopri5). Experiments were executed on a single NVIDIA A100. Metrics are top-1 accuracy (few-shot), OOD accuracy (ImageNet variants), and macro-AUPRC (medical), reported as the mean over three random seeds. Baseline methods include CoOp Zhou et al. (2022), CLIP-Adapter Gao et al. (2021), TIP-Adapter/(f) Zhang et al. (2022), CLAP Silva-Rodríguez et al. (2024), TaskRes(e) Yu et al. (2023), ProKeR Bendou et al. (2025), and LFA Wu et al. (2023).

### 4.1 MULTI- VS. UNI-MODAL ENCODER USAGE ON IMAGENET FEW-SHOT CLASSIFICATION

In this initial experiment, we demonstrate the flexibility and potential advantages in FSF's ability adapt independently trained (uni-modal) text and vision encoders to a few-shot classification task. We focus here on 16-shot ImageNet classification and report similar results on 11 datasets, at a variety of shots, in Sec. A.2 of the appendix. The results are presented in Table 1.

In the left panel we provide results from a variety of methods, in the multi-modal (jointly trained) setting, represented by CLIP with two common possible backbones (RN50 and ViT-B/16), where FSF can be seen to be competitive with leading CLIP adapters. This is consistent with Sec. A.2, where FSF attains the best average accuracy at 16-shots over the entire 11-dataset benchmark, and with Sec. 4.2, where FSF provides superior out-of-distribution accuracies in the multi-modal setting.

Importantly, out of these methods, only LFA and FSF are able to align uni-modal (independently trained) models, with results presented in the right panel. On the text side we include *general-purpose* encoders—GTE-B/L Li et al. (2023e), Qwen-8B Yang et al. (2025), and ARL Liu et al. (2019)—which are not co-trained with images; on the vision side we use MAE-B He et al. (2021), BAM-B Shalam & Korman (2024), and DINOv2-B Oquab et al. (2024).

When text and vision are decoupled, FSF consistently beats LFA across encoder pairs, with larger gains for stronger encoders. CLIP text encoders usually excel on ImageNet, likely due to domain match, but FSF makes other CLIP-free setups competitive: general text encoders (Qwen-8B, GTE-B/L, ARL) with a strong uni-modal vision model (DINOv2-B) reach the same accuracy band as

Table 2: **Out-of-distribution (OOD) 16-shot results on ImageNet variants.** The results are grouped, in between horizontal dashed lines, according to the used encoders (whether multi- or uni-modal). All methods were adapted to (in-distribution) ImageNet and then evaluated on the four (OOD) target distributions. Last row shows, for reference, linear-probe using DinoV2-B on the full ImageNet training set. FSF clearly pushes forward the OOD capabilities of prior works, over the different encoder configurations, even surprisingly surpassing linear probing (in OOD) with an equivalent Vit-B architecture.

| method | text encoder | image encoder | ImageNet | -V2 | -Sketch | -A | -R | avg. OOD |
|---|---|---|---|---|---|---|---|---|
| Zero-Shot | CLIP RN50 | | 60.35 | 51.49 | 33.33 | 21.67 | 55.93 | 40.61 |
| CLIP-A | CLIP RN50 | | 59.02 | 48.15 | 14.63 | 15.75 | 46.29 | 31.21 |
| TIP-A | CLIP RN50 | | 57.81 | 50.32 | 33.59 | 21.88 | 56.98 | 40.69 |
| TIP-A(f) | CLIP RN50 | | 62.27 | 53.99 | 33.75 | 20.47 | 57.22 | 41.36 |
| TaskRes(e) | CLIP RN50 | | 60.85 | 56.47 | 32.80 | 21.28 | 57.93 | 41.29 |
| CLAP | CLIP RN50 | | **65.02** | **56.09** | 34.55 | 21.52 | *59.48* | 42.91 |
| LFA | CLIP RN50 | | 63.88 | 55.79 | 34.37 | **24.31** | 58.13 | *43.15* |
| **FSF**-noOP | CLIP RN50 | | *64.07* | *55.94* | **35.97** | *23.80* | **60.60** | **44.08** |
| **FSF**-GOP | CLIP RN50 | DinoV2_S | 71.07 | 61.83 | 40.40 | 33.97 | 57.83 | 48.51 |
| CLAP | CLIP Vit-B/16 | | **73.38** | *65.00* | *48.35* | 49.53 | *77.26* | 60.04 |
| LFA | CLIP Vit-B/16 | | 72.65 | 64.72 | 48.01 | **51.50** | 76.09 | *60.08* |
| **FSF**-noOP | CLIP Vit-B/16 | | *73.27* | **65.50** | **49.10** | *50.87* | **77.88** | **60.84** |
| **FSF**-GOP | CLIP Vit-B/16 | DinoV2_B | 76.70 | 68.60 | 52.21 | 58.30 | 70.82 | 62.48 |
| Linear probe | DinoV2_B | | **84.50** | 75.1 | 50.60 | 55.10 | 63.30 | 61.02 |

CLIP text. Importantly, by separating modalities, FSF enables replacing both image and text CLIP encoders, clearly improving over CLIP-only results, even at matched ViT-B scale.

## 4.2 OUT OF DISTRIBUTION (OOD) ON IMAGENET VARIANTS

In addition to the in-distribution experiments (Sections 4.1 and A.2), we follow the standard OOD evaluation in FSC, where models are trained on few-shot ImageNet and then evaluated (without further tuning) on four of its variants: ImageNetV2 Recht et al. (2019), ImageNet–Sketch Wang et al. (2019), ImageNet–A Hendrycks et al. (2021a), and ImageNet–R Hendrycks et al. (2021b). Accuracy results appear in Table 2.

There are several important conclusions to be made. First, when comparing between multi-modal methods (the first block of CLIP RN50 and third block of the larger CLIP Vit-B/16), FSA (recall no OP for the multi-modal setting) is highly competitive. On the source Imagenet, it is second to CLAP, but it significantly (and almost consistently) surpasses it in the OOD regime.

Second, moving from the first block of multi-modal CLIP-RN50, by introducing the Dino-V2-S image encoder, hence moving to a uni-modal setting, FSF brings consistent improvements. Likewise, but more significantly, When moving from CLIP Vit-B/16 to the equivalent architecture level DINOv2-B (replacing CLIP's vision tower with an equivalent complexity one), FSF brings further consistent improvements, with accuracy of 76.7 on ImageNet and 62.48 on the OOD variants. We compare these results, that were adapted to Imagenet with only $16,000$ labeled examples (16-shots per class), to those of a standard DINOv2_B linear probe trained on the full ImageNet train split. While FSF is still quite far behind the linear-probe on the source ImageNet, it surpasses it in OOD (on average), which demonstrates how highly robust it is under distribution shift.

## 4.3 MULTI-LABEL CLASSIFICATION ON MEDICAL DATA

Clinical chest X-rays differ materially from the few-shot vision benchmarks used elsewhere in this paper. Images are grayscale with subtle, spatially diffuse cues; labels are *multi-label* (an exam may exhibit several findings), heavily imbalanced (rare pathologies coexist with common ones), and often weak/noisy because supervision is derived from reports. In this regime, threshold-free ranking metrics are preferred: following standard practice for chest X-ray benchmarks, we report macro-AUPRC on VinDr-CXR Nguyen et al. (2020), a seven-finding dataset widely used for evaluation.

Our goal is to test whether FSF can turn a strong *uni-modal* vision encoder into an effective multi-modal classifier by aligning it to an independent text tower—i.e., without pretraining a CLIP-style model on medical data. We therefore use RAD-DINO Pérez-García et al. (2024) as the image back-

Figure 3: **Multi-label classification on VinDr-CXR (7 pathologies; macro-AUPRC).** FSF adaptation results across an increasing support size of $\{16, 32, 64, 128\}$ shots. We experiment with the dedicated RAD-DINO vision encoder, over different text encoders. Results are contrasted with RAD-DINO linear-probe that was trained on entire train set. As can be seen, the usage of CLIP text encoders is inferior to using either the general large-scale Qwen-8B) encoder or the more tailored fine-tuned BiomedCLIP.

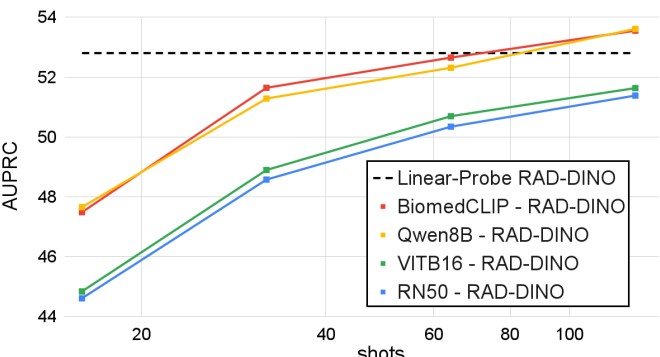

bone (self-supervised on radiology images; not tuned on VinDr-CXR) and pair it with several text encoders spanning medical-specific (BiomedCLIP) and general-purpose (Qwen-8B, CLIP) models. To accommodate the multi-label setting, each training image's label set is converted into a single text target by averaging the normalized label prompts before OP alignment. At inference, we score each class against its OP-aligned prototype.

The results, presented in Fig. 3, are instructive. With RAD-DINO features, FSF paired with the *general* Qwen-8B text encoder tracks BiomedCLIP closely at all shot counts and slightly surpasses it at 128-shot. Notably, both FSF+Qwen-8B and FSF+BiomedCLIP exceed a linear probe trained on the full dataset, despite using only a fraction of the labels, suggesting that FSF's geometry-aware transport regularizes learning under label noise and class imbalance. When we replace RAD-DINO with a generic DINO vision backbone, absolute numbers drop—as expected when leaving the medical domain—but FSF still yields consistent improvements over linear probing across shots (Complete results appear in Tables 6 and 7, Sec. A.4).

Taken together, these results show that FSF can "upgrade" a uni-modal, in-domain vision encoder into a competitive multi-modal classifier by aligning to an independent text encoder. medical-specific text pretraining is helpful but not essential once the visual backbone is in-domain and the alignment/flow is learned. This aligns with our central theme: independently pretrained encoders, connected by OP and refined by flows, can match or even surpass end-to-end multi-modal pretraining in challenging (e.g. OOD and multi-label) settings.

## 5 CONCLUSION

We introduced FSF, a simple and modular framework for few-shot classification that integrates Orthogonal Procrustes alignment with a flow-based alignment prior. This design enables flexible alignment of independently trained vision and text encoders, without requiring large paired corpora or heavy optimization. Extensive experiments demonstrate the value of this capability: FSF consistently improves over the multi-modal baselines across eleven few-shot benchmarks, achieves robust performance under ImageNet distribution shifts, and adapts effectively to domain-specific data such as the VinDr-CXR medical benchmark.

While FSF is effective across diverse settings, it also has limitations. Our method still relies on the availability of strong uni-modal encoders, and the benefits of flow-based alignment may diminish if either side lacks semantic quality. Moreover, although the flow prior is lightweight compared to full model tuning, it still introduces an extra training component beyond closed-form alignment. Future work may explore refined flow designs, scaling to stronger encoders, semi-supervised extensions, or operating directly in token-level latent spaces of images and text.

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

Table 3: **Few-Shot Classification on 11 datasets**. **left**: The standard *multi-modal* CLIP-RN50 setting. FSF is highly competitive with prior works, especially at the higher shot regime. **right**: The *multi-modal* CLIP-RN50 DINOv2-S combination. Three first rows are training-free linear-map alignments and the last two are the FSF with local/global OP alignment, showing the potential of upgrading to a specialized (uni-modal) image encoder. Best/second-best results appear in **bold**/*italics*.

| Method | 1-sh | 2-sh | 4-sh | 8-sh | 16-sh |
|---|---|---|---|---|---|
| Zero-Shot | 57.71 | 57.71 | 57.71 | 57.71 | 57.71 |
| CoOp | 59.56 | 61.78 | 66.99 | 70.11 | 72.53 |
| CLIP-A | 58.43 | 62.46 | 66.18 | 69.87 | 73.35 |
| TIP-A(f) | 60.29 | 62.26 | 65.32 | 68.35 | 71.40 |
| TaskRes(e) | 61.44 | 65.26 | 68.35 | 71.66 | 74.42 |
| CLAP | **62.79** | **66.07** | **69.13** | *72.08* | 74.59 |
| **FSF**-noOP | *62.49* | 65.26 | 68.77 | **72.51** | **75.60** |

| Method | 1-sh | 2-sh | 4-sh | 8-sh | 16-sh |
|---|---|---|---|---|---|
| GOP | 42.40 | 42.40 | 42.40 | 42.40 | 42.40 |
| OP | 52.81 | 59.24 | 64.11 | 67.12 | 69.05 |
| LFA | 50.53 | 60.28 | 67.80 | 73.59 | 78.03 |
| **FSF**-LOP | *54.87* | *62.65* | *70.40* | **76.35** | **80.44** |
| **FSF**-GOP | **59.29** | **65.50** | **70.89** | *75.84* | *79.77* |

# A APPENDIX

## A.1 ALGORITHMS

---

**Algorithm 1** Training

---

**Require:** support set $\Omega = (\{s_i\}_{i=1}^N, \{l_i\}_{i=1}^N)$
**Require:** frozen encoders $f_{\text{img}}, f_{\text{txt}}$
**Require:** initialized flow model $v_\theta$
 1: **compute** image and text embeddings $\{x_i\}_{i=1}^N$ and $\{y_i\}_{i=1}^N$ (Eq. 1)
 2: **estimate** (globally or locally) an OP linear map $W^\star$ (Sec. 3.2)
 3: **align** text to image using OP: $y_i \leftarrow W^\star y_i$ for $i = 1, \ldots, N$
 4: **for** minibatches $(x, y)$ **do**
 5:     **sample** a batch of intermediate times $t \sim \mathcal{U}[0, 1]$
 6:     **obtain** intermediate path points $x_t = \gamma(t; x, y)$
 7:     **compute** target velocities $u(t; x, y)$ (linear or geodesic)
 8:     **update** $\theta$ by minimizing the loss $\mathcal{L}_\rightarrow(\theta)$ (Eq. 5)
 9: **end for**

---

**Algorithm 2** Few-shot inference

---

**Require:** query $s$; class labels $\{l_c\}_{c \in C}$; encoders $f_{\text{img}}, f_{\text{txt}}$; flow model $v_\theta$; map $W^\star$; params $\tau, \alpha$;
 1: **compute** embeddings $x$ and $\{y_c\}_{c \in C}$ (Eq. 1)
 2: **align** text prototypes to image using OP: $y_c \leftarrow W^\star y_c$ for every $c \in C$
 3: **integrate** ODE (Eq. 7) to obtain $z^{(\tau)}$; symmetrically compute $z_c^{(1-\tau)}$;
 4: **compute** class scores $s_c$ (Eq. 9)
 5: **predict** class $c = \arg\max_{c \in C} s_c$

---

## A.2 FEW-SHOT CLASSIFICATION ON 11 DATASETS

We follow the widely adopted CoOp 11-dataset protocol Zhou et al. (2022) under $K \in \{1, 2, 4, 8, 16\}$ shots, providing a diverse testbed covering generic recognition, scenes, textures, remote sensing, actions, and especially fine-grained categories that are particularly challenging for few-shot learning. The benchmarks include ImageNet Deng et al. (2009), StandfordCars Krause et al. (2013), UCF101 Soomro et al. (2012), Caltech101 Fei-Fei et al. (2004), Flowers102 Nilsback & Zisserman (2008), SUN397 Xiao et al. (2010), DTD Cimpoi et al. (2014), EuroSAT Helber et al. (2019), FGVCAircraft Maji et al. (2013), OxfordPets Parkhi et al. (2012), and Food101 Bossard et al. (2014). The detailed top-1 accuracies appear in Table 3.

Table 4: **FSF component ablation on ImageNet (16-shot).** Avg is the mean over shots; green deltas show the incremental gain vs. the previous row. Coupling GOP with the flow stabilizes the low-shot regime.

| OP | flow | 1-shot | 2-shot | 4-shot | 8-shot | 16-shot | Avg. |
|------|----------|--------|--------|--------|--------|---------|-----------------|
| global | w/o | 42.40 | 42.40 | 42.40 | 42.40 | 42.40 | 42.40 |
| local | w/o | 52.81 | 59.24 | 64.11 | 67.12 | 69.05 | 62.47 ↑20.07 |
| local | linear | 55.00 | 62.28 | 69.26 | 73.75 | 78.39 | 67.74 ↑5.27 |
| local | geodesic | 54.87 | 62.65 | 70.40 | 76.35 | 80.44 | 68.94 ↑1.20 |
| global | geodesic | 59.29 | 65.50 | 70.89 | 75.84 | 79.77 | 70.26 ↑1.32 |

The left panel reports results for the standard *multi-modal* CLIP-RN50 encoder. Recall that since CLIP's image–text space is already co-trained we do not apply OP in this setting. The trend here is consistent with previous observations that learned priors benefit from richer supervision: prompt/logit calibration can be slightly favored at 1–2 shots due to its tight inductive bias, while FSF's non-linear, geometry-aware transport yields consistently growing improvements as the number of shots increases.

The right panel reports results for the *uni-modal* combination of a CLIP-RN50 text encoder and a DINOv2-S image encoder. When text and image encoders are trained independently, alignment becomes the main challenge. A global OP, fitted once on 1M CC3M pairs, provides a very initial baseline of 42.40, which is still inferior to the baseline Zero-Shot (but well aligned) CLIP, despite using a stronger vision encoder. Computing local OP on the support set increasingly improves the alignment as the number of shots grows. LFA, improves over the local OP, especially at the higher shots, due to applying iterative updates on the linear map. However, FSF is able to significantly boost these results, due to the flow-based non-linear alignment. Consistently throughout our results, and as was discussed in Sec. 3.2, global/local OP is better suited for the lower/higher shot regimes.

Importantly, the capability of using any independent combination of uni-modal encoders is shared only by FSF and LFA among the compared methods. As is clear in this case, and was made more so in the other experiments, this flexibility can be highly beneficial, due to the ability to choose an adequate combination of encoders for a given task.

### A.3 ABLATIONS

#### A.3.1 FSF COMPONENTS.

In Table 4 we ablate *alignment* (OP: local vs. global/GOP) and the *flow prior* on ImageNet across shots. Conclusions: *(i) Geometry matters.* Using a geodesic (spherical) path instead of a linear one consistently helps once a few labels are available, raising the shot-averaged score (68.94 vs. 67.74), which aligns with our unit-normalized embeddings. *(ii) GOP as a low-shot prior.* Although the global $W^\star$ is fitted on an *external* dataset (subset of CC3M) and starts from a weaker OP than local OP, coupling GOP with the flow stabilize the low-shot regime: about +4.4 at 1-shot and +2.9 at 2-shot over local + geodesic, while staying within 1 point at 8/16; it achieves the highest average across shots (70.26). This suggests the learned flow is *more robust* to alignment quality and compensates residual cross-modal mismatch. *(iii) Net effect.* Moving from local OP (no flow) to local + geodesic yields $\sim +6.5$ on average; adding GOP contributes a further $\sim +1.3$. In practice, prefer GOP for $K \leq 2$, use local OP at higher shots, and adopt geodesic flow by default.

#### A.3.2 HYPER-PARAMETERS.

We examine the sensitivity of the mixing weight $\alpha$ in Eq. 9 and the integration horizon $\tau$ (Sec. 3). Recall our selection protocol from Sec. 4: $\alpha$ and $\tau$ are tuned only on at most $\max(K, 4)$ shots from the official validation splits and then fixed. Here we visualize accuracy surfaces on the test sets *for analysis only*—no test-time selection.

**Figure layout.** Fig. 4 shows two heatmaps and a compact trend summary: **(a)** CLIP text + DINOv2 image (independent encoders) and **(b)** CLIP-RN50 (co-trained encoders). Each heatmap has two rows (16-shot on top, 1-shot below) and three columns (ImageNet, Oxford Pets, DTD). Rows sweep

Table 5: **Residual MLP ablation on ImageNet 16-shot (DINOv2-B + Qwen-8B).** Top-1 (%). Best in **bold**; defaults are underlined.

| Depth | 1 | 2 | 4 | 8 |
|---|---|---|---|---|
| Acc. (%) | **76.47** | **76.47** | 76.37 | 76.17 |
| **Width** | 512 | 1536 | 2048 | 4096 |
| Acc. (%) | 75.83 | 76.37 | 76.47 | **76.73** |
| **time_dim** | 128 | 256 | 512 | 1024 |
| Acc. (%) | 76.30 | 76.37 | 76.43 | **76.47** |

Table 6: **VinDr-CXR (RAD-DINO vision).** Macro-AUPRC (%). Best per row in **bold**.

| Shots | Linear Probe | FSF-BiomedCLIP | FSF-Qwen8B | FSF-CLIP ViT-B/16 | FSF-CLIP RN50 |
|---|---|---|---|---|---|
| 16 | **52.80** | 47.48 | 47.65 | 44.83 | 44.60 |
| 32 | **52.80** | 51.64 | 51.28 | 48.89 | 48.57 |
| 64 | **52.80** | 52.65 | 52.31 | 50.69 | 50.34 |
| 128 | 52.80 | 53.55 | **53.61** | 51.63 | 51.38 |

$\tau \in [0, 1]$, columns sweep $\alpha \in [0, 1]$; lighter is better. Edge cases: $\alpha=0$ uses only the learned adapters (at $\tau=1$ this reduces to the image→text adapter, at $\tau=0$ to the text→image adapter). Conversely, $\alpha=1$ falls back to the base model (zero-shot CLIP in (b); OP-only in (a)). Panel **(c)** summarizes the mean optimal $\alpha$ and $\tau$ vs. shots over five datasets (ImageNet/DTD/Oxford Pets/Oxford Flowers/SUN397).

**What changes with more data.** Three consistent patterns emerge. *(i) Robustness grows with $K$.* From 1-shot to 16-shot, bright plateaus expand and the surfaces become smooth, indicating reduced sensitivity to $(\alpha, \tau)$; wide ranges perform near-optimally once a few labels are available. *(ii) Less reliance on the base model.* The preferred $\alpha$ decreases with shots in both panels and is systematically smaller for CLIP+DINOv2 than for CLIP (panel **(c)**): roughly $\alpha \approx 0.28 \rightarrow 0.10$ (CLIP+DINOv2) vs. $0.85 \rightarrow 0.45$ (CLIP) from 1→16 shots, showing that the learned flow contributes increasingly as supervision grows, especially with independent encoders. *(iii) Mild shift in transport.* The preferred $\tau$ moves modestly lower with shots in both settings (about $0.50 \rightarrow 0.35$ for CLIP+DINOv2; $0.45 \rightarrow 0.30$ for CLIP), giving relatively more weight to the text→image adapter side.

**Practical recipe.** FSF does *not* require brittle parameter tuning: for quick deployment we find *CLIP-only* robust in $\alpha \in [0.4, 0.7]$, $\tau \in [0.3, 0.5]$, and *CLIP+DINOv2* robust in $\alpha \in [0.05, 0.3]$, $\tau \in [0.35, 0.55]$, with both decreasing slightly as $K$ increases. These settings align with prior practice while underscoring that FSF remains strong even without careful per-dataset tuning.

### A.3.3 RESIDUAL MLP ARCHITECTURE

Our velocity network is a residual MLP (SiLU) with time conditioning. We ablate *depth*, *width*, and *time_dim* on ImageNet (16-shot) using **DINOv2-B** for vision and **Qwen-8B** for text, with geodesic flows and OP alignment fixed. Results in Table 5 (Top-1, %) report a 4 value sweep over the depth, width and time_dim parameters, where default values are underlined and are used for the independent sweeping of each other two parameters. It shows small variance ($\leq 0.34$ pp) per row, indicating that FSF is stable to reasonable architectural choices. A shallower/wider model yields a slight gain, but our default $4 \times 1536$ with time_dim $= 256$ strikes a good accuracy/latency trade-off and is used throughout.

### A.4 FULL RESULTS ON VINDR-CXR (MULTI-LABEL)

We report macro-AUPRC (%) on VinDr-CXR (7 pathologies) for $K \in \{16, 32, 64, 128\}$ shots in T. FSF averages the positive label prompts *before* OP alignment and learns geodesic flows. Linear probe uses the full training set for the given vision backbone. Complete results appear in Tables 6 and 7.

Table 7: **VinDr-CXR (generic DINO vision).** Macro-AUPRC (%). Best per row in **bold**.

| Shots | DINO Linear Probe | FSF-BiomedCLIP |
|---|---|---|
| 16 | 31.60 | **33.05** |
| 32 | 31.60 | **33.07** |
| 64 | 31.60 | **34.81** |
| 128 | 31.60 | **37.38** |

## A.5 VISUALIZING OP AND FLOWS ON OXFORD-PETS (QUALITATIVE)

To build intuition for FSF with *independent* encoders, we visualize in A.5 how OP and the learned flow reshape text embeddings toward image features on a single Oxford-Pets class ("Abyssinian", 16 shots). Text is encoded by a generic BERT Devlin et al. (2019) tower and images by the CLIP image encoder (no joint training). On the *left*, we show the 16-shot support set for the class. In the *center*, we render what the *text→image* embeddings "look like" along the integration path by training a diffusion decoder (DiT Peebles & Xie (2022)) conditioned on CLIP *image* features; we then feed the transported text-in-image features at $t \in \{0, 0.5, 1\}$ to synthesize indicative images. The goal is not realism, but qualitative interpretability of the trajectory. On the *right*, we plot a spherical (unit-norm) PCA of image features with OP-projected text prototypes and the learned flow field (gray arrows). Circles denote image features, squares the image features after flow, and triangles the text prototypes; points are colored by class.

Two patterns emerge. First, **LOP vs. GOP initialization**: LOP (fitted on the episode's support) places text prototypes closer to the class manifold than GOP (estimated once from external data), which appears farther and more diffuse in feature space. Second, **the flow matters**: in the center panels, LOP at $t=0$ can decode to dogs rather than the intended cat, but as we integrate to $t=0.5$ and $t=1.0$ the synthesis becomes increasingly cat-like; GOP starts even less informative (near gray-scale), yet the same flow progressively steers it to the correct semantics. Together, these views illustrate why OP provides a geometry-preserving bridge while the learned flow performs the heavy lifting to resolve residual cross-modal mismatch.

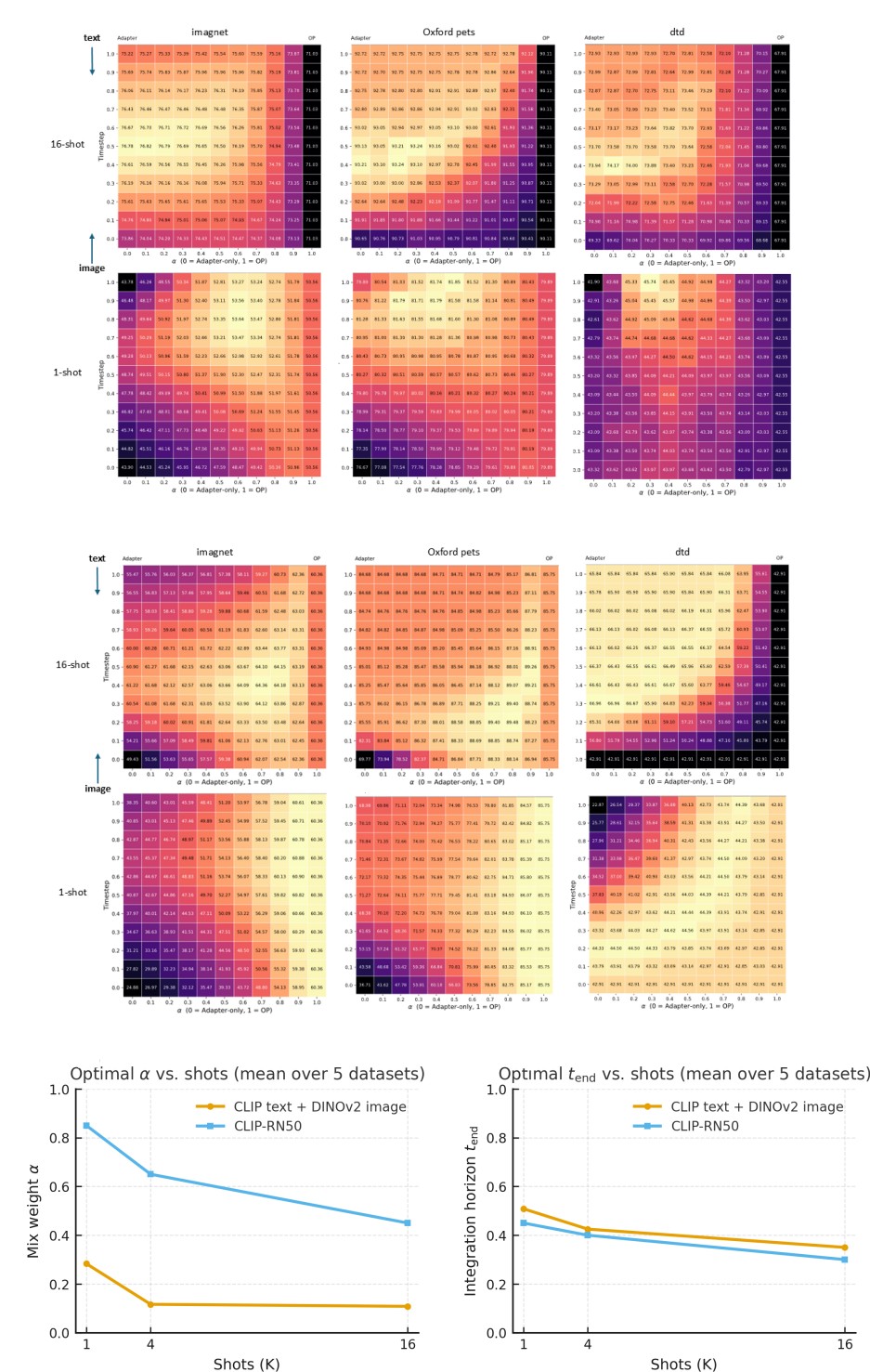

Figure 4: **FSF hyper-parameter space and trends. top and middle are results for DINO and CLIP respectively**. These are heatmaps over $(\alpha, \tau)$ for 1-shot (bottom row) and 16-shot (top row); lighter is better. Edge cases: $\alpha{=}0$ uses only adapters (at $\tau{=}1$ image$\rightarrow$text; at $\tau{=}0$ text$\rightarrow$image). $\alpha{=}1$ falls back to the base model - zero-shot CLIP in CLIP (top) and OP-only in DINO (bottom). **(c)** Summary of optimal $\alpha$ and $\tau$ vs. shots (mean over INet/DTD/Pets/Flowers/SUN397). As shots increase, $\alpha$ decreases—especially with independently pretrained encoders (CLIP+DINOv2)—and $\tau$ shifts lower and smooths out, indicating reduced sensitivity and strong performance without tuning.

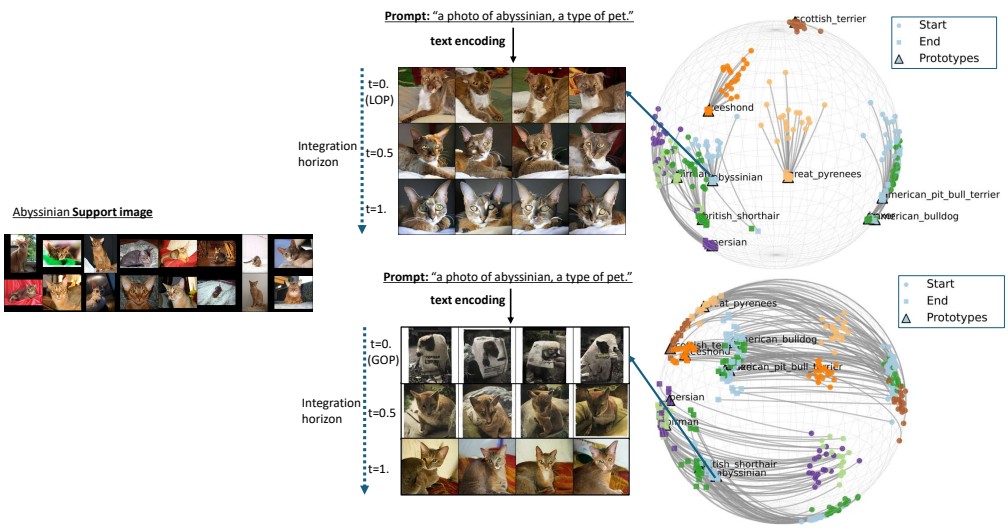

Figure 5: **Qualitative visualization of OP and flow on Oxford-Pets ("Abyssinian").** Left: class support (16 shots). Center: DiT-based visualizations of the text→image trajectory at $t \in \{0, 0.5, 1\}$ (top: LOP+Flow; bottom: GOP+Flow). Right: spherical PCA of image features with OP-projected text prototypes and the learned flow (gray). LOP initializes closer to the class manifold; GOP is farther due to global fitting on external data. In both cases, integrating the flow moves text embeddings toward the correct visual semantics, underscoring the necessity of flow on top of OP for independently trained encoders. **The figure is high-resolution. Please zoom in digitally for a best possible view.**

