# OpenReview forum: "Flow-Based Alignment of Uni-Modal Vision and Text Encoders for Few-Shot Image Classification"
_ICLR.cc/2026/Conference — ICLR 2026 Conference Desk Rejected Submission_

### Official Review · Reviewer_Ws4H · 2025-10-28

**Soundness:** 3
**Presentation:** 3
**Contribution:** 3
**Rating:** 8
**Confidence:** 5

**Summary:**

The paper proposes **FSF** (Few-shot-Flow) which is a framework that aligns arbitrary uni-modal vision and text encoders for cross-modal **FSC**, enabling the use of stronger or domain-specific models without join pretraining. **FSF** combines two components: **Orthogonal Procrustes (OP) Alignment** and **Flow-Matching Prior**. Unlike CLIP-dependent adapters, **FSF** bridges independent encoders without large paired datasets or heavy training. The proposed method demonstrates consistent improvements across 11 benchmarks, demonstrating robustness to distribution shifts.

**Strengths:**

1) The proposed method achieved consistent performance via a novel combination of Orthogonal Procrustes (OP) alignment, a closed-form linear mapping that preserves geometric structure, and a lightweight flow-matching prior that models non-linear cross-modal transport in latent space. The study addresses a gap in adapting uni-modal models for few-shot classification without large paired datasets.

2) The study provides a flexible alignment of uni-modal encoders, which can be extend to other multi-modal tasks, particularly in resource-constrained or specialized domains.

3) **FSF** consistently achieves competitive accuracy compared to SOT few-shot classification methods on standard benchmarks even when using the standard multi-modal CLIP backbone.

**Weaknesses:**

1) The paper claims the flow "lightweight", but lacking report of training time, inference latency compared to baselines.

2) The paper is lacking a fair comparison with other non-linear mapping between feature spaces such as Optimal Transport (OT) and Gromov-Wasserstein (GW) methods.

**Questions:**

1) Could you provide a report of training and inference time compared to baselines.

2) Could you conduct experiments for comparing the proposed method with other non-linear mapping such as Optimal Transport or Gromov-Wasserstein?

---

> ### Author Response · Authors · 2025-11-22
> **Response**
>
> We thank the reviewer for the comments and insightful questions. Below we address the main points:
>
> **Q1**: "Report training time and inference latency."
>
> **A1**: We thank the reviewer for this request that lead to an informative analysis (Result 4). We benchmarked FSF against common CLIP-based approaches and found that our method shows attractive trade-offs between quality and efficiency. For instance, CLIP-Adapter requires 50 minutes of training to reach 61.33\% accuracy. In contrast, our 4-layer FSF achieves a significantly higher 64.20\% accuracy in just 17 minutes. This makes FSF nearly 3x faster to train while being more accurate. Furthermore, the table highlights FSF's good scalability: our lightest depth-1 model already outperforms all baselines (63.67\% accuracy) with just 13 minutes of training, demonstrating that users can scale model capacity to gain additional performance at negligible cost. We attribute this speed to our flow-matching objective, which learns from direct (image-text) pairs, avoiding the computationally expensive batch-wide pairwise similarity computations used by other methods. Finally, this efficiency extends to test time: even with the (Dopri5) ODE solver (which typically uses only around 5 steps), our inference speed (11.8 ms) per-query is practically identical to that of the baselines.
>
> **Q2**: "comparison with other non-linear mapping methods between feature spaces such as Optimal Transport (OT) and Gromov-Wasserstein (GW)."
>
> **A2**: To our understanding, although there are many theoretical connections between these tools and flow-matching, we find the OT and GW frameworks less relevant to this specific supervised adaptation setting. We will explain our understanding, but please let us know if you have other particular ideas in mind.
>
> OT and GW are primarily designed for unsupervised alignment (i.e., discovering unknown correspondences between sets of features), whereas in our few-shot setting, the offline learning stage is fully supervised since the support-set image-to-label pairings are given. Unlike flow-matching, that uses these correspondences to learn a continuous flow-field that can be applied to unobserved test-time data points, OT and GW are typically used (and are computationally efficient) in discrete cases. In this sense, we don't see how they could be provide the equivalent functionality of FM in this setting.
>
> We can imagine how OT or GW could be used, independently to our alignment, as a replacement to the simple cosine-distance based class assignments. Such a contribution would be independent to ours and while it could be attractive, it relies on the setting being 'transductive' - meaning that it assumes that all queries can be jointly processed, and that they have some predetermined spread (e.g. uniform) between classes.

---

### Official Review · Reviewer_5BWx · 2025-10-29

**Soundness:** 4
**Presentation:** 3
**Contribution:** 3
**Rating:** 8
**Confidence:** 3

**Summary:**

This paper presents FSF, a novel framework for few-shot image classification that enables flexible alignment of uni-modal (independently trained) vision and language encoders. The method first applies a closed-form Orthogonal Procrustes (OP) linear map to establish an initial feature space alignment. It then learns a lightweight flow-matching prior to model non-linear transformations between image features and text prototypes. Extensive experiments show that FSF outperforms baselines on standard few-shot benchmarks, excels under distribution shifts, and adapts effectively to domain-specific tasks like medical imaging, all while maintaining efficiency.

**Strengths:**

- This paper is well-written and easy to follow.
- The proposed orthogonal procrustes and lightweight flow-matching are novel and effective.
- The experimental results demonstrate the effectiveness of the proposed methods.

**Weaknesses:**

- Missing comparison with recent methods. The authors only compare FSF with early baselines and do not make a comparison with recent methods such as GDA-CLIP[1], TIMO[2]. Could the authors provide a justification?
- Format error in line 442.



Reference:

[1] A Hard-to-Beat Baseline for Training-free CLIP-based Adaptation. ICLR24

[2] Text and Image Are Mutually Beneficial: Enhancing Training-Free Few-Shot Classification with CLIP. AAAI25

**Questions:**

- Justification of baseline choices as mentioned in the weakness section.
- I'm kind of surprised by the results of the flow matching training could be done with such limited data. Could the authors provide more insights into this phenomenon?

---

> ### Author Response · Authors · 2025-11-22
> **Response**
>
> We thank the reviewer for the comments and insightful questions. Below we address the main points:
>
> **Q1**: "Missing comparison with recent methods (GDA-CLIP, TIMO). Could the authors provide a justification?"
>
> **A1**: We thank the reviewer for pointing out these very relevant baselines. Regarding the justification for our original selection, we initially prioritized comparisons with methods that address cross-modal alignment (like LFA, ICCV'23) or ones that represent the recent state-of-the-art in adaptation (such as CLAP, CVPR'24 and ProKer, CVPR'25). Our goal was to frame FSF primarily around its unique ability to leverage strong, independent uni-modal encoders (like DINOv2 or RAD-DINO) - a critical flexibility that CLIP-specific methods like GDA-CLIP and TIMO inherently lack.
>
> Nevertheless, we have now benchmarked FSF against these suggested methods using the standard CLIP-RN50 backbone (Result 1). The results confirm that FSF remains state-of-the-art even in this restricted (multi-modal) setting: it achieves the highest OOD generalization (Avg. OOD 44.08\%), surpassing both TIMO (43.22\%) and GDA-CLIP (41.26\%). This support our claim that FSF offers general robustness while possessing the unique advantage of modular flexibility that these baselines cannot provide.
>
> **Q2**: Surprised by flow matching training with such limited data. Provide insights.
>
> **A2**: This is an important observation, which touches on a fundamental aspect of our design. The key reason behind FSF's data-efficiency lies in the distinctions between the (discriminative) task it needs to solve and the more standard (generative) ones that flow matching typically handles. Generative models (like Stable Diffusion) face the difficulty of mapping high-entropy Gaussian noise to complex pixel distributions, requiring massive datasets to cover the data manifold. In contrast, FSF learns a much simpler transport map between two robust and semantically rich embedding spaces (text and vision). Because the encoders are frozen and pre-trained, the "heavy lifting" of feature extraction is already done. Further, the task is paired (and hence allows a simple application of rectified flow), in the sense of learning fixed geodesic paths between support image-text pairs (as opposed to denoising stochastic noise), providing a stable, low-variance learning signal. Lastly, the alignment needs to be tailored to the specific few-shot instance in hand, whose complexity is very small, compared to the need to align an entire space in detail. Albeit the deterministic nature of the task, the continuous time parameter $t$ acts as a geometric data augmentation, exposing the network during training to a continuous trajectory of interpolations rather than just fixed endpoints, effectively regularizing the model against overfitting, even with limited shots.

---

> ### Comment · Reviewer_5BWx · 2025-11-27
>
> Thanks to the author for the valuable insights. My concerns are addressed, and I will raise my rating to 10.

---

### Official Review · Reviewer_dW7u · 2025-10-31

**Soundness:** 3
**Presentation:** 4
**Contribution:** 3
**Rating:** 6
**Confidence:** 4

**Summary:**

The paper proposes a flow-based model for multimodal few-shot classification. The approach is highly interesting, thoughtfully addresses various experimental settings, and is well-supported by comprehensive experiments that align closely with the proposed method.

**Strengths:**

1. The paper innovatively employs a flow-based model to achieve multimodal alignment under few-shot settings. The lightweight flow-matching framework appears to incur minimal computational overhead during training, yet yields significant performance gains.
2. The experimental evaluation is exceptionally thorough: the main text includes results on ImageNet and medical CXR datasets, while the appendix extends the analysis to multiple additional small-scale datasets. Moreover, the authors explore a wide variety of model combinations, offering robust evidence for the effectiveness of their method.
3. The writing is clear and accessible, making the technical content easy to follow.

**Weaknesses:**

**Major Weakness:**

I do not identify any significant weaknesses in the paper.

**Minor Weaknesses:**

1. For independently trained unimodal models, what performance would be achieved if FSF were applied directly without OP? The paper does not report this ablation, although I can intuitively sense the function of this component from the author's description.
2. The appendix includes a comparison between GOP and LOP , but it does not explain why each variant excels in different experimental settings. A deeper analysis of this behavior would be insightful.
3. In Table 3 (right panel), the second row appears to correspond to LOP but is mistakenly labeled as “OP.”
4. Line 250 contains a typo: “Sec.” should be formatted properly (e.g., “Section”).
5. The heading of Section 4.3 suffers from formatting issues.
6. Throughout many tables, CLIP model names (e.g., “RN50”) are used to refer specifically to their text encoders. While this is generally understandable, it can momentarily confuse readers who naturally associate these names with the full CLIP model.

**Questions:**

1. I encourage the authors to address the minor weaknesses above, particularly by adding the suggested ablation studies and clarifying the GOP vs. LOP behavior.
2. Section 4.3 presents additional experiments in the medical domain, which, while useful, seem less central than the GOP/LOP analysis and ablation studies in the appendix. Consider swapping their placement to prioritize the more fundamental methodological insights.
3. Would applying OP to standard CLIP (without FSF) lead to performance improvements? This would help isolate the contribution of OP itself.
4. Including t-SNE visualizations of feature embeddings before and after flow-based alignment would offer intuitive, qualitative evidence of the method’s impact.
5. The paper focuses exclusively on few-shot classification. However, I am curious whether the proposed flow-based alignment could be scaled to large-scale multimodal datasets (e.g., LAION-400M). For instance, initializing with DINOv2 and CLIP’s text encoder, then training the flow model for cross-modal alignment. Could this yield representations that surpass standard CLIP?

Although I have raised several questions, I consider this a high-quality paper. I am inclined to give it a score of 7–8. However, since the current scoring system cancels a 7, I am provisionally assigning a 6. If the authors can address a substantial portion of my concerns, I would be happy to raise my score to 8 or higher.

---

> ### Author Response · Authors · 2025-11-22
> **Response - Part 1**
>
> We thank the reviewer for the comments and insightful questions. Below we address the main points:
>
> ---
> **Q1**: "..what performance would be achieved if FSF were applied directly without OP?"
>
> **A1**: We thank the reviewer for raising this question and for suggesting a respective ablation that could highlight the importance of the OP module. To isolate its effect, we ran the requested experiment (w. vs. w/o OP) using a strong uni-modal pair (CLIP-L/14 text and DINOv2-B image) where the feature dimensions match (otherwise OP or some dimensionality reduction is strictly required). The table (Result 3) shows that removing OP leads to a 4.1-point drop in 1-shot accuracy and the gap narrows as the number of shots increases, which is expected. In the low-shot regime, the flow model does not have enough data to learn the cross-modal alignment from scratch, so the closed-form OP initialization provides the needed stability. With more shots (e.g. 16), supervision becomes strong enough for the flow model to learn the mapping on its own, although OP remains beneficial throughout.
>
>
> ---
> **Q2**: "Would applying OP to standard CLIP (without FSF) lead to performance improvements?"
>
> **A2**: CLIP was explicitly trained on large-scale image-text data, in order to produce well-aligned multi-modal representations, so it is not expected to benefit from a linear correction estimated from a very limited number of support examples. To verify this intuition, and to address the reviewer's question, we ran a new ablation (Result 3) that applies OP to both zero-shot CLIP and our FSF model (with the same multi-modal CLIP). On average, applying OP leads to large performance drops: 20.09 points for CLIP and 8.68 points for FSF. For both models, the negative effect is strongest in the low-shot regime, where OP overfits the small support set and disrupts the underlying alignment. The effect becomes weaker as more shots are available, although applying OP does not surpass the baseline's original performance. These findings indicate that OP alone is not a reliable adaptation mechanism. For multi-modal models, we therefore omit OP and solely rely on the more expressive non-linear flow-matching prior.
>
>
> ---
> **Q3**: "The appendix includes a comparison between GOP and LOP, but it does not explain why each variant excels in different experimental settings. A deeper analysis of this behavior would be insightful."
>
> **A3**: Comparisons between Global OP (GOP) and Local OP appear in several locations. Direct comparisons (per number of epochs) appeared in Tables 3 and 4  (now 4 and 5) of the appendix, in the PCA projections of the latent space in Figure 6 and in the updated manuscript - also in the new Figure 5 which compares the alignment trajectories in t-SNE space.
>
> We interpret the performance trade-off between GOP and LOP as a standard bias-variance effect in few-shot alignment. GOP acts as a stable, low-variance prior: because it is computed once on a large external dataset (CC3M), it provides a reliable initialization that is not affected by the noise of the small support set, which explains why it performs better in the very low-shot setting (1-2 shots). In contrast, LOP fits the alignment directly to the current support set. This gives lower bias and more task-specific adaptation, but it also means higher variance when data is scarce, which can lead to overfitting. As the number of shots increases (e.g., 8-16), the support set becomes a better estimate of the target task, and in this regime LOP's flexibility becomes more helpful than GOP's stability, leading to better results. This explains why each variant excels in different conditions and supports our practical recommendation: use the global prior (GOP) when data is very limited, and switch to the local version (LOP) once more supervision is available.

---

> > ### Author Response · Authors · 2025-11-22
> > **Response - Part 2**
> >
> > **Q4**: "Including t-SNE visualizations of feature embeddings before and after flow-based alignment would offer intuitive, qualitative evidence of the method's impact."
> >
> > **A4**: We thank the reviewer for this valuable suggestion. Visualizations of the alignment (both OP and flow) in latent space is important for qualitative evaluation of the method. We already provided an illustration of our flow trajectories (was Fig. 5 in Appendix A.5 and now is Fig. 6 in C.2). However, it is spherical-PCA based and focuses on single class, from a generative perspective.
> >
> > We therefore added a new Section C.1 and Figure 5, with the requested t-SNE projections, before and after flow, in two different complete FSC instances, including a comparison between the LOP and GOP linear initializations. These support the empirical result, regarding the qualities of the alignment initializations (either LOP or GOP) and the following flow trajectories.
> >
> >
> > ---
> > **Q5**: "Could flow-based alignment be scaled to large-scale multi-modal datasets (e.g., LAION-400M)?"
> >
> > **A5**: We thank the reviewer for this proposal, which aligns with our own roadmap for future work. We believe that scaling FSF to large datasets like LAION-400M using strong, frozen uni-modal experts (e.g., DINOv2 and Qwen) is indeed a highly promising direction. Although our current work focuses on few-shot discriminative features, there is clear evidence from the generative domain (e.g. Stable Diffusion 3) that flow-matching models scale very well with data and we believe that a “Foundation FSF” trained at such a scale could learn a robust, generalized transport map that could have potential interesting usages in wider discriminative tasks (e.g. image-captioning, visual question-answering, retrieval) and even generative ones, for which our method has shown very initial results (Fig. 5, now Fig.6, of the appendix).

---

> > > ### Comment · Reviewer_dW7u · 2025-11-23
> > > **Thank you for the response.**
> > >
> > > I've raised my score to 10. In my view, after the revisions, this paper is clearly at Oral presentation quality.

---

### Official Review · Reviewer_eWWC · 2025-11-03

**Soundness:** 3
**Presentation:** 3
**Contribution:** 3
**Rating:** 4
**Confidence:** 3

**Summary:**

This paper introduces a novel framework for few-shot image classification that can leverage arbitrary, independently pre-trained uni-modal vision and text encoders. The key idea is to bridge the gap between these separate encoders in a two-step process. First, it applies a closed-form Orthogonal Procrustes (OP) mapping to achieve a geometry-preserving linear alignment of the text and image embedding spaces. Second, it trains a lightweight, continuous-time flow model that learns a non-linear transport path between the aligned embeddings. Experiments across multiple benchmarks (ImageNet, OOD variants and VinDr-CXR) demonstrate that FSF is highly effective, often outperforming methods reliant on jointly trained multi-modal encoders.

**Strengths:**

- The primary strength of FSF is its ability to decouple the choice of encoders from the alignment process. Unlike methods that are constrained to work with jointly trained vision-language models like CLIP, FSF can flexibly combine arbitrary uni-modal encoders. This is a significant advantage, as it allows ones to plug in state-of-the-art vision or text models that may be more powerful or better suited for a specific domain (e.g., vision encoder for chest X-rays), as demonstrated by the strong results with RAD-DINO.

**Weaknesses:**

- While the proposed alignment method is interesting, the motivation of using it for few-shot image classification is unclear. Why do we need to align pretrained image and text encoders for few-shot classification?

- The few-shot ImageNet experiment could be flawed. The classes or even images in the ImageNet dataset are likely to be covered by image encoder pretraining dataset. The authors are encouraged to consider the following few-shot learning benchmark.

A Broader Study of Cross-Domain Few-Shot Learning. ECCV 2020

- More recent baselines should be compared because LFA was published in 2023.

Frozen Feature Augmentation for Few‑Shot Image Classification. CVPR 2024

Multi-Label Few-Shot Image Classification via Pairwise Feature Augmentation and Flexible Prompt Learning. AAAI 2025

**Questions:**

Increased Hyperparameter Complexity: FSF introduces several new hyperparameters compared to simpler alignment methods, including the intermediate integration time τ and the mixing parameter α for inference. While the appendix shows these can be tuned for good performance, this adds a layer of complexity and potential tuning cost that might be a barrier for easy adoption.

---

> ### Author Response · Authors · 2025-11-22
> **Response - Part 1**
>
> We thank the reviewer for the comments and insightful questions. Below we address the main points:
>
> **Q1**: "..the motivation of using it (FSF) for few-shot image classification is unclear. Why do we need to align pretrained image and text encoders for few-shot classification?"
>
> **A1**:
> We appreciate the opportunity to clarify this point.
>
> In few-shot classification with image–text models, the class name is turned into a text embedding that acts as the prototype for that class, and each query image is classified by comparing its embedding to these prototypes. This comparison only works if image and text embeddings lie in a common feature space, which is naturally the case when using multi-modal encoders, such as CLIP. In such multi-modal scenarios, our alignment is only optional (and many alternative approaches exist, such as adapter-based models) and when using FSF, we do not apply the OP stage (as we empirically justify in Result 3), but our flow-based alignment was shown to be beneficial in all of the experiments that consider multi-modal encoders (see Tables 1-3 in the paper).
>
> More importantly, our main motivation is to encourage and support the use of stronger or domain-specific uni-modal (independently pretrained) encoders (e.g., DINOv2 for images and Qwen for text), whose latent spaces are unrelated (and perhaps of different dimensions), in which case cosine (or other type) similarities are not meaningful. When using FSF, the Orthogonal Procrustes map provides the initial linear alignment needed to place image and text features in a shared space, and the flow adds a non-linear correction that improves stability in the few-shot regime. In the uni-modal setting, the alignment step is not optional - it is what enables few-shot classification when the encoders were never co-trained. The merits of working in the uni-modal setting (enabled by FSF) are shown across all of the main experiments, and perhaps especially on the medical data experiment (Sec. 4.3) as an example of data for which multi-modal encoders do not exist (and would be hard to obtain due to scarcity of particular paired image-text data).
>
> **Q2**: "The few-shot ImageNet experiment could be flawed. The classes or even images in the ImageNet dataset are likely to be covered by image encoder pretraining dataset. The authors are encouraged to consider the following few-shot learning benchmark."
>
> **A2**: We understand this concern but wish to clarify that our setup exactly follows the common practice in modern few-shot classification (FSC) work, which is based on adapting pretrained foundation encoders, that were trained on large-scale data without enforcing strict separation from the evaluation sets, as we detail in the following.
>
> In the vision-only FSC literature, many works follow this approach. “Rethinking Few-Shot Image Classification: A Good Embedding Is All You Need?” (ECCV 20) and more recent works such as “Towards Few-Shot Adaptation of Foundation Models” (ICLR 24) reflect what has become the main trend in recent years - utilizing strong frozen encoders and focusing mainly on adaptation mechanisms. The same pattern is well established in vision-language FSC, mostly relying on frozen CLIP backbones (e.g. CLIP-Adapter ICCV'21 and CoOp IJCV'22), but also combining unimodal vision transformers (e.g. CaFo CVPR'23 and AMU-Tuning CVPR'24). These methods use frozen vision encoders like MAE, MoCo, and DINO - all pretrained without labels (some on large-scale data) - and evaluate them directly on few-shot ImageNet without requiring strict disjointness between pretraining and evaluation images.
>
> In fact, we argue that few-shot methods benefit from using encoders that are not only strong but also domain-relevant (e.g., DINOv2 for natural images, RAD-DINO for medical data). FSF is designed precisely to support such flexibility, which we view as one of our key contributions.

---

> > ### Author Response · Authors · 2025-11-22
> > **Response - Part 2**
> >
> > **Q3**: "..consider the following few-shot learning benchmark (ECCV'20). More recent baselines should be compared (CVPR'24, AAAI'25) because LFA was published in 2023"
> >
> > **A3**: We thank the reviewer for broadening the scope with these references. Within the standard "foundation model adaptation" protocol, we already compare against recent state-of-the-art methods such as CLAP (CVPR'24) and ProKeR (CVPR'25), both published after LFA (2023). The request for additional recent baselines was also raised by Reviewer R3, who suggested GDA-CLIP (ICLR'24) and TIMO (AAAI'25), which we have added both to our evaluation and show (Result 1) that FSF outperforms these directly comparable baselines in OOD generalization.
> >
> > The remaining suggested works (ECCV'20, CVPR'24, AAAI'25) operate under different experimental protocols. The ECCV'20 benchmark uses Cross-Domain Few-Shot Learning, which requires meta-training on a source domain (e.g., mini-ImageNet). In contrast, FSF follows the CoOp / CLIP-Adapter protocol, which assumes no upstream meta-training and evaluates frozen pretrained encoders directly. The CVPR'24 and AAAI'25 works also address tasks (e.g., vision-only augmentation, multi-label classification) that are outside the standard single-label V-L few-shot setting that we study in this work.
> >
> >
> > **Q4**: "Increased Hyperparameter Complexity ($\tau, \alpha$) as a barrier to adoption."
> >
> > **A4**: We understand the reviewer's concern regarding potential tuning costs. In practice, however, these parameters do not present a barrier to adoption. The mixing parameter $\alpha$ is used in many recent few-shot adapters (e.g., Tip-Adapter, CLIP-Adapter), where it serves the same purpose of balancing prior knowledge with task-specific adaptation, so FSF is consistent with such common practice.
> >
> > Moreover, our sensitivity study (Appendix A.3.2, Fig. 4) shows that FSF is highly robust: both $\tau$ and $\alpha$ exhibit wide performance plateaus, and broad ranges of values yield near-optimal accuracy. To further reduce tuning effort, we provided a simple “deployment-ready” configuration in the end of the appendix section A.2.2 (e.g., fixing $\tau = 0.4$, $\alpha = 0.6$ for CLIP backbones) that performs well across benchmarks without per-dataset adjustment.

---

### Author Response · Authors · 2025-11-22
**General response**

We sincerely thank all four reviewers for their detailed, constructive, and encouraging feedback and are delighted that they recognize the value of our work. We are particularly grateful for the insightful inquiries, regarding computational cost, baseline comparisons, and component necessity, which we believe have enabled us to provide a strengthened version of the manuscript.

Since the questions are mostly distinct, we address them in separate per-reviewer responses. We wish to mention here, that we have provided the following new experimental results which were requested by the different reviewers. We added these results to the manuscript, but also present and discuss them shortly in the following response blocks (and more in length in the per-reviewer replies):

1. **Comparison with recent SOTA** (**R1** and **R3**): Extension of the "Out-of-distribution (OOD) on ImageNet variants" experiment (Table 2 of the paper) with GDA-CLIP (ICLR'24) and TIMO (AAAI'25).
2. **OP usage ablation in the uni-modal setting** (**R2**): Evaluation of FSF without OP on independently trained encoders.
3. **OP usage ablation in multi-modal setting** (**R2**): Evaluation of OP applied to standard CLIP and to CLIP-based FSF.
4. **Cost-Benefit Analysis** (**R4**): Detailed benchmarking of training time and inference latency against standard CLIP-based approaches.
5. **OP and Flow Visualization** (**R2**): Multi-class t-SNE plots of OP alignment and flow trajectories.

These results indicate that FSF provides consistent improvements in both efficiency and performance compared to existing baselines. Our cost-benefit analysis (**Result 4**) reveals that FSF trains roughly 3x faster than CLIP-Adapter, and achieves higher accuracies, with negligible increase in inference latency. Our comparison with the recent TIMO and GDA-CLIP methods (**Result 1**) shows superior  Out-Of-Distribution (OOD) generalization accuracy. Our ablations regarding the use of Orthogonal Procrustes (OP) confirm our main design choices: We show on one hand (**Result 2**) that OP is a structural necessity for uni-modal alignment but on the other hand (**Result 3**) that it is destructive when naively applied to pre-aligned multi-modal models. Finally, the visualizations (**Result 5**) of the local and global OP alignment initializations and the following flow trajectories, demonstrate additional qualities of our solution and differences between several of its design options.

---

> ### Author Response · Authors · 2025-11-22
> **Summary of new results (requested experiments) - Part 1**
>
> **Result 1**: Comparison with recent SOTA (**R1** and **R3**)
> Extension of the "Out-of-distribution (OOD) on ImageNet variants" experiment (Table 2 of the paper) with GDA-CLIP (ICLR'24) and TIMO (AAAI'25).
>
> | method      | INet  | INet-V2 | INet-Sk | INet-A | INet-R | avg. OOD |
> |-------------|-------|---------|--------|--------|--------|----------|
> | GDA-CLIP    | *64.13* | 55.67  | 34.32 | 19.72 | 55.30 | 41.26   |
> | TIMO        | **64.63** | **56.40** | *35.96* | *22.06* | *58.47* | *43.22* |
> | **FSF** | 64.07 | *55.94* | **35.97** | **23.80** | **60.60** | **44.08** |
>
> -**Interpretation**:
> FSF is on-par with the recent TIMO and GDA-CLIP (suggested by **R3**) on  ImageNet (to which the methods were adapted) but clearly outperforms them on Out-Of-Distribution (OOD) generalization, with average accuracy of 44.08\%. Best/second-best results are in bold/italics.
>
> ---
>
> **Result 2**:  OP usage ablation in the uni-modal setting (**R2**)
> Evaluation of FSF without OP (Orthogonal Procrustes) on independently trained encoders.
>
> | method        | 1-shot | 2-shot | 4-shot | 8-shot | 16-shot | avg.  |
> |---------------|--------|--------|--------|--------|---------|-------|
> | **FSF** (w/o OP)  | 56.63  | 64.60  | 70.07  | 73.47  | 75.55   | 68.06 |
> | **FSF** (w/ OP) | **60.73** | **67.47** | **72.17** | **75.00** | **77.00** | **70.47** |
> | *gain*        | +4.10  | +2.87  | +2.10  | +1.53  | +1.45   | +2.41 |
>
> -**Interpretation**: This ablation highlights the importance of the OP module in the uni-modal setting. We ran FSF on ImageNet, using a strong uni-modal pair (CLIP-L/14 text and DINOv2-B image) with matching feature dimensions (otherwise OP or some dimensionality reduction are anyway strictly required). The results show that FSF can be entirely flow-based and work without the OP initial alignment, but this comes at a significant average cost of 2.41 in accuracy, with an expected widening of the gap towards the low-shot regime in which the flow model does not have enough data to learn the cross-modal alignment from scratch.

---

> > ### Author Response · Authors · 2025-11-22
> > **Summary of new results (requested experiments) - Part 2**
> >
> > **Result 3**:  OP usage ablation in multi-modal setting (**R2**)
> > Evaluation of OP applied to standard CLIP and to CLIP-based FSF.
> >
> > | method              | 1-shot | 2-shot | 4-shot | 8-shot | 16-shot | avg.  |
> > |---------------------|--------|--------|--------|--------|---------|-------|
> > | ZS CLIP (w/o OP) | **60.33** | **60.33** | **60.33** | **60.33** | **60.33** | **60.33** |
> > | ZS CLIP (w/ OP)     | 33.27  | 37.78  | 41.36  | 43.61  | 45.21   | 40.24 |
> > | *gain*              | -27.06 | -22.55 | -18.97 | -16.72 | -15.12  | -20.09 |
> > |                     |        |        |        |        |         |       |
> > | **FSF** (w/o OP)     | **60.90** | **61.40** | **62.00** | **63.17** | **64.13** | **62.32** |
> > | **FSF** (w/ OP)         | 35.90  | 43.07  | 61.97  | 63.13  | **64.17** | 53.64 |
> > | *gain*              | -25.00 | -18.33 | -0.03  | -0.04  | 0.04    | -8.68 |
> >
> > -**Interpretation**: The jointly trained (multi-modal) CLIP-B text and image encoders are well aligned and as can be seen in the top zero-shot ImageNet CLIP experiment - applying the linear OP correction, especially when based on a limited number of support examples, has a significant negative impact on the alignment. Similar behavior is observed when applying such OP corrections in the FSF setting, prior to the flow alignment, with negative impact especially in the low-shot regime. This ablation justifies our decision to work without OP in multi-modal settings.
> >
> > ---
> > **Result 4**:  Cost-Benefit Analysis (**R4**)
> > Detailed benchmarking of training time and inference latency against standard CLIP-based approaches.
> >
> > | model              | train time | trainable params | inference speed (ms) | accuracy (%) |
> > |--------------------|------------|------------------|------------------|--------------|
> > | Zero-shot CLIP     | 0          | 0                | 10.2             | 55.41        |
> > | Linear Probe CLIP  | 13min      | 1.02M            | -                | 53.44        |
> > | CoOp               | 14h 40min  | 0.02M            | 299.6            | 60.46        |
> > | CLIP-Adapter       | 50min      | 0.52M            | 10.6             | 61.33        |
> > |         |          |            |                  |                  |
> > | **FSF** (depth=1)  | 13min      | 4.37M            | 11.4             | 63.67        |
> > | **FSF** (depth=2)  | 15min      | 6.93M            | 11.5             | 63.73        |
> > | **FSF** (depth=4)  | 17min      | 12.05M           | 11.8             | 64.20        |
> >
> > -**Interpretation**: This new cost analysis on 16-shot ImageNet, with all methods using the multi-modal CLIP-B encoder, reveals that FSF shows strong efficiency-quality tradeoffs. For example, the largest depth-4 variant achieves state-of-the-art accuracy of 64.20\% while training in just 17 minutes (roughly 3x faster than CLIP-Adapter's 50 minutes), while achieving higher accuracy and maintaining comparable inference latency (+1.6 ms). Further efficiency can be gained (with a gracefull decrease in performance) by reducing the flow MLP network depth by factors of 2 or 4. Note that the aparent higher number of parameters used by FSF has negligible effect on the total model size which is dominated by the image and text encoders (~150M in this case).
> >
> > ---
> > **Result 5**:  OP and Flow Visualization (**R2**)
> > Multi-class t-SNE plots of OP alignment and flow trajectories, which qualitatively demonstrate the qualities of the alignment initializations (either LOP or GOP) and the following flow trajectories.
> >
> > -**Results and interpretations appear in Appendix C.1 of the revised manuscript.**

---

### Note · Program_Chairs · 2026-01-17
**Submission Desk Rejected by Program Chairs**

The following references in this submission do not refer to real documents and/or have major errors in bibliographic information:

 Yuke Zhou, Yixuan Yang, Shuyang Wang, Ying Li, Wanli Zhang, Baining Ni, and Vincent Wong. Coop: Context optimization for prompt learning. International Journal of Computer Vision (IJCV), 2022.

Ziyao Zhang, Renrui Zhang, Hongsheng Li, Ping Luo, and Peng Gao. Cafo: Cross-modal adaptation by feature optimization for few-shot clip. In IEEE/CVF Conference on Computer Vision and Pattern Recognition (CVPR), 2023.

Xiang Li, Shuang Li, Weitao Wan, Guangrun Wang, and Limin Wang. Geodesic multi-modal mixup for robust fine-tuning. In Advances in Neural Information Processing Systems (NeurIPS), 2023c.